# The Over-Expression of Two *R2R3-MYB* Genes, *PdMYB2R089* and *PdMYB2R151*, Increases the Drought-Resistant Capacity of Transgenic *Arabidopsis*

**DOI:** 10.3390/ijms241713466

**Published:** 2023-08-30

**Authors:** Xueli Zhang, Haoran Wang, Ying Chen, Minren Huang, Sheng Zhu

**Affiliations:** 1State Key Laboratory of Tree Genetics and Breeding, Ministry of Education of China, Co-Innovation Center for the Sustainable Forestry in Southern China, Nanjing Forestry University, Nanjing 210037, China; zhangxueli@njfu.edu.cn (X.Z.); ychen@njfu.edu.cn (Y.C.); mrhuang@njfu.edu.cn (M.H.); 2Jiangsu Key Laboratory for the Research and Utilization of Plant Resources, Institute of Botany, Jiangsu Province and Chinese Academy of Sciences, Nanjing Botanical Garden, Memorial Sun Yat-Sen, Nanjing 210014, China; njlydxwhr@163.com; 3College of Biology and the Environment, Nanjing Forestry University, Nanjing 210037, China

**Keywords:** poplar, *R2R3-MYB* genes, drought-responsive, stomatal closure

## Abstract

The *R2R3-MYB* genes in plants play an essential role in the drought-responsive signaling pathway. Plenty of *R2R3-MYB* S21 and S22 subgroup genes in *Arabidopsis* have been implicated in dehydration conditions, yet few have been covered in terms of the role of the S21 and S22 subgroup genes in poplar under drought. *PdMYB2R089* and *PdMYB2R151* genes, respectively belonging to the S21 and S22 subgroups of NL895 (*Populus deltoides* × *P. euramericana* cv. ‘Nanlin895′), were selected based on the previous expression analysis of poplar *R2R3-MYB* genes that are responsive to dehydration. The regulatory functions of two target genes in plant responses to drought stress were studied and speculated through the genetic transformation of *Arabidopsis thaliana*. *PdMYB2R089* and *PdMYB2R151* could promote the closure of stomata in leaves, lessen the production of malondialdehyde (MDA), enhance the activity of the peroxidase (POD) enzyme, and shorten the life cycle of transgenic plants, in part owing to their similar conserved domains. Moreover, *PdMYB2R089* could strengthen root length and lateral root growth. These results suggest that *PdMYB2R089* and *PdMYB2R151* genes might have the potential to improve drought adaptability in plants. In addition, *PdMYB2R151* could significantly improve the seed germination rate of transgenic *Arabidopsis*, but *PdMYB2R089 could* not. This finding provides a clue for the subsequent functional dissection of S21 and S22 subgroup genes in poplar that is responsive to drought.

## 1. Introduction

As an important factor restricting the sustainable forest-based industry and economy, drought is a serious issue for the growth and development of forest trees across the world. Studies have shown that, in the past decade, trees in China’s Three-North Shelterbelt have suffered extensive degradation and death [1]. In addition to natural aging, drought and drought-induced pests or diseases are considered to be important external factors that accelerate tree death [1,2]. Plants respond to water deficit by changing their morphological architecture, expressing drought-resistant genes, and synthesizing osmoregulation substances and phytohormones [3,4,5,6]. Transcription factors (TFs), as potential candidate genes in drought-tolerant breeding, play an essential regulating role in plant response to drought stress [7,8,9]. With more than 100 species, *Populus* is one of the most widely distributed and adaptable tree species in the world. Recent research on poplar drought-responsive TFs (e.g., WRKY, NAC, MYB, bZIP, and AP2/ERF) has made some progress, which provides a reference value for other tree species in response to drought stress [10].

The *R2R3-MYB* gene family, as a *MYB* gene subfamily with the largest number of members, is widely involved in cell differentiation, cell cycle regulation, plant development and metabolism, phytohormone, and abiotic stress response [11,12,13,14,15]. Previous reports of dehydration-responsive R2R3-MYB TFs focused primarily on herbs, represented by *Arabidopsis thaliana*. For example, *AtMYB30*, *AtMYB60*, *AtMYB94*, and *AtMYB96* were found to be involved in biological processes that respond to abscisic acid (ABA) and drought signals and then participated in drought response by regulating stomatal movement and root growth [16,17,18,19]. In addition, both *AtMYB94* and *AtMYB96* could specifically bind to the MYB-binding sequence (MBS) in the promoter region of genes related to wax biosynthesis (e.g., *WSD1* and *KCS2*), and directly regulate the expression of the genes involved in epidermal wax biosynthesis under drought conditions, thus resulting in plants with elevated drought resistance [20,21]. In contrast, only a few poplar R2R3-MYB TFs had been documented as being responsive to drought stress. For example, *P. tomentosa PtoMYB170* of *P. tomentosa* could regulate stomatal closure to control water loss and regulate lignin accumulation during poplar wood formation, which led to secondary cell wall thickening [22]. *P. simonii* × *P. nigra PsnMYB108* had the ability to regulate the scavenging of reactive oxygen species (ROS) [23]. *P. trichocarpa PtrMYB94* improved the tolerance of transgenic plants to water shortage by regulating the expression of ABA and drought stress-related genes (e.g., *ABA1*, *DREB2B*, and *P5CS2*) [24]. It has been shown that PtoMYB142 could directly bind to the promoters of wax biosynthesis genes (e.g., *CER4* and *KCS6*) and induce their expression to increase wax accumulation in poplar leaves, thus resulting in elevated drought resistance [25]. However, there is still a lack of research on the regulatory mechanism of poplar *R2R3-MYB* genes against drought stress. Thereby, the study of the drought-responsive mechanism of R2R3-MYB TFs should have great significance for the exploration of genes that improve stress resistance, and this will provide a theoretical basis for improving drought-tolerant traits in poplar.

There were 126 gene members of the *R2R3-MYB* gene family in *A. thaliana*, which were divided into 24 subgroups [26], and more gene members (about 210) in *P. trichocarpa* were divided into 23 subgroups [23,27]. The *R2R3-MYB* genes involved in drought stress have been studied and reported in many of the R2R3-MYB subgroups of *A. thaliana*, but the studies on poplar are very limited. For instance, the subgroup S01 gene of *A. thaliana* responds to drought stress by regulating stomatal movement and root growth [16,17,18,19]; the *R2R3-MYB* gene member of subgroup S04 and S14 responded to ABA/drought signals and is involved in ABA-mediated seed germination [28,29,30]; the subgroup S09 genes are involved in wax biosynthesis under drought stress [31]. The *R2R3-MYB* genes of the subgroup 21 (S21) and 22 (S22) are closely related in their evolutionary relationship, and a number of their genes have been shown to be implicated in plant response to drought stress [27]. For example, *AtMYB52* belongs to the S21 subgroup, and its overexpression conferred ABA hypersensitivity during post-germination growth and increased the drought tolerance of seedlings [32]. As an *AtMYB52* homologous gene, *PtrMYB2R089* (Potri.007G134500, also named *PtrMYB161*) was shown to bind to its upstream TFs to increase the number of xylem vessels and reduce the growth of plants, and these traits are normally related to adaptive to drought stress [33]. On the other hand, *A. thaliana* R2R3-MYB subgroup S22 is composed of *AtMYB44*, *AtMYB77*, *AtMYB73,* and *AtMYB70*. *AtMYB44* could positively regulate ABA signaling to induce stomatal closure, thus conferring drought/salt tolerance in *Arabidopsis* [34]. *AtMYB77* participated in drought-induced lateral root growth by promoting nitric oxide (NO) synthesis [35]. *AtMYB73* participated in plant drought stress by regulating seedling root length, stomatal closure, and the seed germination rate [36]. In our previous study, we found that the S21 and S22 subgroups of the poplar *R2R3-MYB* gene family had a relatively close genetic relationship, and their gene members had similar dehydration-responsive expression patterns [27]. *PdMYB2R089* (Potri.007G1345001) and *PdMYB2R151* (Potri.014g0225001) of *P. deltoides* × *P. euramericana* cv. ‘Nanlin895′ are homologous genes of *AtMYB52* and *AtMYB73*, respectively. The expression levels in the roots were significantly higher than in stems and leaves [27]. In addition, the *PdMYB2R089*/*151* gene exhibited an unimodal gene expression pattern in leaves and roots under drought stress, indicating that it was possibly induced by drought stress [27].

In order to study the function of poplar S21 and S22 subgroup genes in plant drought response, we selected one gene from each subgroup for functional characterization. In this study, *PdMYB2R089* (Potri.007G134500.1, S21) and *PdMYB2R151* (Potri.014G022500.1, S22) were selected for prior bioinformatics analyses, including phylogenetic analysis, physicochemical properties, gene structure, and protein–protein interaction prediction. Subsequently, transgenic *Arabidopsis* plants that heterogeneously overexpressed the *PdMYB2R089*/*151* genes were obtained by using the *Agrobacterium*-mediated floral dip method. Further, we analyzed the effects of *PdMYB2R089*/*151* on root growth, stomatal movement, seed germination, the various physiological indexes under drought stress, and the life cycle of transgenic plants. The research regarding the function of *PdMYB2R089* and *PdMYB2R151* genes under drought stress might be significant for the study of the functional and regulatory mechanisms of poplar *R2R3-MYB* S21 and S22 subgroup genes.

## 2. Results

### 2.1. Phylogenetic Analysis and Physicochemical Properties of PdMYB2R089 and PdMYB2R151 Proteins

In this study, we constructed phylogenetic trees for the S21 and S22 subgroup gene members in *P. trichocarpa* and *A. thaliana*. *PdMYB2R089* and *PdMYB2R151* were closely related to the homologous genes *PtrMYB2R089* and *PtrMYB2R151* in *P. trichocarpa* and to *AtMYB52* and *AtMYB73* in *A. thaliana* (Figure 1a). Meanwhile, the interspecific evolutionary tree showed that *PdMYB2R089*/*151* had the closest genetic relationships to that of other *Populus* species, followed by the *Salix*, which is a sister genera of *Populus*. In addition, they were also closely related to other woody plants, while they were related to herbs such as *A. thaliana*, *Oryza sativa*, and *Triticum aestivum* (Figure 1b).

The *PdMYB2R089* gene was 744 bp, encoding 247 amino acids (AAs); the *PdMYB2R151* gene was 936 bp, encoding 311 AAs. All proteins contained 20 kinds of AAs, and the higher content of AAs was Leu, Ser, Arg, Glu, and Pro (Appendix A). The isoelectric points (pI) of PdMYB2R089 and PdMYB2R151 were 8.74 and 9.02, respectively. The aliphatic index reflects the thermal stability of proteins. The aliphatic index for the two genes were 60.81 and 76.21, respectively, indicating that the two PdMYB2R proteins had little difference in terms of thermal stability. In addition, the grand average of hydropathicity (GRAVY) of the two proteins was lower than 0, showing that the two proteins were hydrophilic proteins. It was consistent with the results of homologous genes *PtrMYB2R089*/*151* in *P. trichocarpa* [27].

### 2.2. Analysis of Secondary Structure and Tertiary Structure of PdMYB2R089 and PdMYB2R151 Protein

The protein secondary structure prediction indicated that the two PdMYB2R proteins were mainly composed of alpha-helices and random coils, with a small amount represented by beta turns (Appendix A). The AAs forming alpha-helices and random coils accounted for about 55% and 30% of the total AAs, respectively, while the beta turns accounted for about 5%. Further, their tertiary structure was formed by the winding of the polypeptide chain based on the secondary structure (Appendix A). We could still see the conformation formed by the secondary structure from the tertiary structure of the protein, and most regions had an alpha-helix and random curling conformation.

The hydrophobicity of the AAs reflected protein folding, and this played an important role in maintaining the structure of the biofilms. Therefore, the hydrophilicity and hydrophobicity of the two proteins were studied in detail. The hydrophilic regions of the PdMYB2R089/151 proteins were mainly in the N-terminal, whereas the hydrophobic regions were mainly in the C-terminal (Appendix A). In general, the number of hydrophilic AAs (score < 0) was significantly higher than that of the hydrophobic AAs (score > 0). As an aside, the result indicated the absence of transmembrane domains on the PdMYB2R089 and PdMYB2R151 proteins (Appendix A).

### 2.3. Function Domain Analysis and Protein–Protein Interaction Prediction

The functional domains of the PdMYB2R089 and PdMYB2R151 proteins, with their homologous proteins in *Populus* (e.g., *P. trichocarpa*, *P. tomentosa*, and *P. euphratica*), *Salix* (e.g., *S. suchowensis* and *S. purpurea*), and *A. thaliana*, were analyzed. As shown in Figure 2, the N-terminal of the proteins all contained an MYB DNA-binding domain (DBD), which accounted for about one-third to one-quarter of the total length of the PdMYB2R protein sequences, and the MYB DBD of PdMYB2R089 and PdMYB2R151 contained two tandem incomplete replicates (R2 and R3), each consisting of approximately 50 AAs and formed three helices, with the domains in the N-terminal of the two proteins conserved (Figure 2 and Appendix A). The SANT/MYB domain at the N-terminal was typical of R2R3-MYB-type TFs and had DNA-binding functions, which is consistent with the results of previous studies [37]. Additionally, we also found that PdMYB2R089 and PdMYB2R151 TFs contained regions of variable length and low sequence conservation at the C-terminal [38].

We predicted the proteins that interacted with PdMYB2R089 and PdMYB2R151, respectively. The results showed that PdMYB2R151 could interact with PtrEXLB3/4, PtpRR10/11, WRKY, and ARFs; PdMYB2R089 mainly interacted with ARF TFs and beta-1,4-xylosyltransferase irx14 (Figure 3a). Previous studies had shown that the gene members of EXL, PRR, WRKY, and ARF regulate plant response to drought stress [39,40,41,42]. As the protein object that both proteins may interact with, ARFs have been shown to regulate root growth by interacting with MYB77 TF, a member of the S22 subgroup [43]. When taken together, these results suggest that PdMYB2R089/151 may interact with some drought-related proteins in response to drought stress.

Our previous study demonstrated that PdMYB2R151 was a nuclear-localized protein [27]. In this study, we added the functional localization analysis of the PdMYB2R089 protein. The subcellular localization assay showed that 35S::GFP had the fluorescent signal in the whole protoplast, whereas PdMYB2R089 only exhibited the fluorescent signal in the nucleus and not in the cytoplasm or cell membrane (Figure 3b). The result showed that PdMYB2R089 is also a nuclear localization protein.

### 2.4. Function of PdMYB2R089 and PdMYB2R151 Gene under Drought Stress

Previous research has showed that *AtMYB52* is associated with the regulation of cell wall biosynthesis and plant growth under drought stress [32]. As a homologous gene of *AtMYB52*, *PtrMYB2R089* was documented to function as a genetic and epigenetic regulatory switch, controlling cell wall component biosynthesis, growth, and adaptation [33]. Members of the *R2R3-MYB* gene family (e.g., *AtMYB44*, *AtMYB73*, *AtMYB70*, *AtMYB77*) in the S22 subgroup of *A. thaliana* are commonly involved in abiotic stress. *PdMYB2R0151* is a homologous gene of *AtMYB73*, belonging to the S22 subgroup of the poplar R2R3-MYB family. Our previous study found that *PdMYB2R089* and *PdMYB2R151* were induced in roots, stems, and leaves by drought stress [27]. In order to understand the regulatory function of *PdMYB2R089* under drought stress, a transgenic *Arabidopsis* strain that heterogeneously overexpresses (OEs) *PdMYB2R089* and *PdMYB2R151* was obtained, respectively. Through iterative screening, homozygote transgenic plants without trait separation in terms of progeny were finally obtained (Appendix A).

#### 2.4.1. Overexpressed *PdMYB2R089* and *PdMYB2R151* Affected the Growth of Plants under Drought Stress

At day 0 of stress treatment, *PdMYB2R089*-OE *Arabidopsis* had an above-ground and root growth advantage compared to the wild-type (WT) *Arabidopsis* (Figure 4a,b), but *PdMYB2R151*-OE did not (Figure 5a,b). Under 0% FEG6000 stress, the above-ground growth of *PdMYB2R089*-OE *Arabidopsis* was slightly greater than WT, and the roots were slightly longer than the WT plants. Under drought stress, especially 10% PEG6000 treatment, the above-ground and root growth of the WT plants was severely inhibited. Meanwhile, the inhibition degree of the *PdMYB2R089*-OE transgenic plants was significantly lesser than that of the WT. *PdMYB2R089*-OE transgenic *Arabidopsis* had more lateral roots and longer roots than the WT plants under drought stress (Figure 4b). Our previous study discovered that the expression level of *PdMYB2R089* in roots was particularly significant when compared with other tissues, which was equivalent to 290 times that of the leaves [27], indicating its regulatory role in plant roots. However, the effects of *PdMYB2R151* on above-ground and root development were only slightly better than those of WT *Arabidopsis* but not particularly significant (Figure 5).

Then, the total fresh weight (FW) and root fresh weight (root-FW) of the WT and transgenic plants were measured on the 6th day of drought stress. As shown in Figure 6a, the FW of the *PdMYB2R089*-OE and *PdMYB2R151*-OE transgenic plants was always significantly higher than that of the WT plants under different conditions (0%, 5%, and 10%). With an increase in PEG6000 concentration, the FW of both the WT and *PdMYB2R089*-OE transgenic *Arabidopsis* showed a decreasing trend, but the decreasing range of transgenic plants was significantly less than that of the WT. Nevertheless, the different drought stress (5% and 10%) treatments did not seem to cause FW changes in the *PdMYB2R151*-OE transgenic plants (Figure 6a). In addition, under different stress treatment conditions, the root-FW of *PdMYB2089*-OE *Arabidopsis* was significantly higher than that of the WT plants. However, there was no significant difference between the root-FW of *PdMYB2R151*-OE *Arabidopsis* and that of the WT under drought stress (5%, 10%), which was consistent with the root phenotype results above.

#### 2.4.2. Overexpressed *PdMYB2R089* and *PdMYB2R151* Reduced Plant Drought Damage

Based on the *PdMYB2R089/151*-OE transgenic plants, several plant physiological indexes were determined, including malondialdehyde (MDA), soluble protein, and peroxidase (POD) activity. The results showed that the content of MDA in the WT was significantly higher than that in *PdMYB2R089*-OE and *PdMYB2R151*-OE transgenic plants after a week of water shortage (Figure 6b), indicating that the degree of damage to transgenic plants under drought stress was significantly lower than that of the WT. On the other hand, although there was no significant difference in soluble protein between the WT and transgenic plants, the POD activity of *PdMYB2R089*-OE and *PdMYB2R151*-OE transgenic plants was significantly higher than that of the WT plants. The results suggested that *PdMYB2R089* and *PdMYB2R151* could reduce the production of harmful substances in plants by enhancing the activity of antioxidant enzymes, thus enhancing the drought tolerance of plants.

#### 2.4.3. Overexpressed *PdMYB2R089* and *PdMYB2R151* Promoted Stomatal Movement under Drought Stress

In order to study the regulatory function of the *PdMYB2R089* and *PdMYB2R151* genes on stomatal movement under drought stress, we compared the stomatal pore size of the WT and transgenic plant leaves after drought treatment and rehydration. Under normal water conditions, the stomatal size of the WT and *PdMYB2R089*-OE *Arabidopsis* showed practical unanimity (Figure 7a), but the stomatal size of *PdMYB2R151*-OE was smaller than that of the WT plants (Figure 7b). After drought stress treatment, the degree of stomatal closure of the *PdMYB2R089*-OE and *PdMYB2R151*-OE transgenic plants was more significant than that of WT *Arabidopsis*, and the stomatal conductance of transgenic *Arabidopsis* was restored after rehydration treatment, whereas that of WT *Arabidopsis* was not fully restored to what it was before drought-stress treatment. The results suggested that the overexpression of *PdMYB2R089* and *PdMYB2R151* could promote stomatal movement and might reduce water loss in plants under drought stress, thus enhancing the drought tolerance of the plants.

#### 2.4.4. Overexpressed *PdMYB2R151* Promoted Seed Germination

The research on seed germination under different treatment conditions showed that when under a 0% and 5% PEG6000 concentration, the germination rate of the *PdMYB2R151*-OE transgenic seeds reached 100%, which was significantly higher than that of the WT; under a 10% PEG6000 concentration, the germination rate of the *PdMYB2R151*-OE transgenic seeds was still higher than that of the WT (Appendix A). These results suggested that the overexpression of *PdMYB2R151* could promote *Arabidopsis* seed germination. Moreover, *PdMYB2R089* might have no significant effect on seed germination (Appendix A).

### 2.5. Effect of PdMYB2R089 and PdMYB2R151 on the Flowering and Fruiting Time of Plants

Plant flowering is crucial for the transformation from vegetative to reproductive growth [44]. The protein–protein interaction prediction showed that PdMYB2R089 and PdMYB2R151 might interact with PRR-protein-influencing plant flowering [45] (Figure 3a). In the study, the WT and transgenic *Arabidopsis* seedlings were transplanted into the soil at the same time to observe their flowering and fruiting time under the same condition. The results showed that *PdMYB2R089*-OE *Arabidopsis* began to bolt about 15 days after transplanting, flowered at about 19 days, and fruited at about 26 days, while the WT *Arabidopsis* only began bolting at about 26 days (Figure 8a). *PdMYB2R151*-OE transgenic *Arabidopsis* began to bolt about 18 days after transplanting, flowered at about 24 days, and fruited at about 26 days (Figure 8b). The results indicated that overexpression of *PdMYB2R089 and PdMYB2R089151* could induce earlier flowering and fruiting and shorten the growth cycle of the plants.

## 3. Discussion

Our results suggest a close relationship between the poplar R2R3-MYB S21 and S22 subgroups. Many studies have shown that the genes belonging to the R2R3-MYB S21 and S22 subgroups play important regulatory roles in *Arabidopsis* response to drought stress [34]. Therefore, we selected *PdMYB2R089* (S21) and *PdMYB2R151* (S22) genes in light of our previous study. The results showed that PdMYB2R089 and PdMYB2R151 may have high thermal stability and similar conserved domains. It implies that the R2R3-MYB protein recognizes similar DNA-binding sequences and may share some similarities in function [46]. Subsequently, we predicted the proteins that might interact with the PdMYB2R089/151 proteins and found that the proteins interacting with PdMYB2R089/151 were mainly ARF transcription factors. Various studies have revealed that ARFs play important roles in regulating drought and salinity stress responses in plants [42,47]. For instance, *IbARF5* could increase ABA, proline, and superoxide dismutase (SOD) activity and decrease H_2_O_2_ content, which helps transgenic *Arabidopsis* to increase its tolerance to salt and drought [48]. In addition, among the PdMYB2R151 interacting proteins, the EXLB and PRR proteins have also been reported to be involved in plant drought stress. The genes encoding PRR proteins play significant roles in plant environmental stresses and circadian clocks. For example, *Capsicum annuum CaPRR2*-silenced pepper plants exhibited an altered stomatal conductance and elevated MDA contents, which indicated that *CaPRR2* negatively regulates tolerance to drought and salt stress [49]. These results suggest that PdMYB2R089 and PdMYB2R151 may interact with these drought-related proteins to participate in drought stress by regulating plant growth and stomatal movement and may affect plant drought tolerance through antioxidant stress.

Previous reports have suggested that *AtMYB52* (S21 subgroup) might be related to ABA signal transduction pathways and secondary wall deposition [50]. Therefore, it has been proposed that the changes in cell wall structure caused by *AtMYB52* overexpression may trigger ABA hypersensitivity or *AtMYB52* may directly affect ABA metabolism and response [51]. As an *AtMYB52* homologous poplar gene, *PtrMYB2R089* (Potri.007G134500) has been reported to have a variety of regulatory functions related to wood cell wall formation, growth, and adaptation [33], And we found that PdMYB2R089 may interact with glycosyltransferases (beta-1,4-xylosyltransferase irx14), which are involved in the synthesis of glucuronoxylan hemicellulose in secondary cell walls. It was suggested that *PdMYB2R089* is a potential regulator that affects normal growth and development homeostasis. Therefore, we conducted an in-depth study on the regulatory role of *PdMYB2R089* under drought stress and found that this gene is a positive regulatory factor of drought by regulating plant root growth, stomatal closure, and antioxidant stress. These results indicate that *PdMYB2R089*, like *AtMYB52*, is involved not only in secondary wall deposition but also in the drought stress response process in plants. Our results once again verify the close relationship between wood formation and the ABA signal transduction pathway [33,51].

Previous reports have shown that four members (e.g., *AtMYB44*, *AtMYB70*, *AtMYB73*, and *AtMYB77*) of the *Arabidopsis* S22 subgroup were implicated in root growth and development [50]. For example, it was found that the ABA receptor coding gene PYL8 directly interacts with transcription factors AtMYB77, AtMYB44, and AtMYB73 to enhance the auxin signaling pathway and promote the growth of lateral roots [52]. *AtMYB70* could also regulate root development and seed germination through ABA-auxin signaling pathways [53]. However, *PdMYB2R151*-OE *Arabidopsis* showed slight above-ground and root growth advantages under 10% PEG drought stress, but this was not particularly obvious. The effect of this gene on plant roots under drought stress and its regulatory mechanism still need to be further studied. Jung et al. (2007) found that overexpression of *AtMYB44* could reduce the expression of the *PP2C* phosphatase gene and then positively regulate ABA to induce stomatal closure and enhanced drought tolerance in *A. thaliana* [54]. We found that *PdMYB2R151* genes could induce leaf stomatal closure under soil water shortage, suggesting that *PdMYB2R151* may also enhance plant drought tolerance by reducing water loss in plants.

Dehydration can cause the excessive production of reactive ROS in plants, thus disturbing the balance between the production and clearance of ROS [55]. Subsequently, ROS accumulation in plants causes toxic MDA, ultimately leading to oxidative stress [56]. The antioxidant enzymes (POD, SOD, and CAT) produced in plants can effectively remove ROS and reduce oxidative damage [57]. Our result showed that PdMYB2R089 and PdMYB2R151 may interact with ARFs, and ARF transcription factors have been reported to be involved in the drought-induced oxidative stress pathways in plants [48]. PdMYB2R089 and PdMYB2R151 could enhance POD activity and reduce the damage of MDA to *Arabidopsis*, indicating that PdMYB2R089 and PdMYB2R151 enhanced drought tolerance by activating the ROS-scavenging enzyme system and played a positive role in the oxidative stress caused by drought. Notably, PdMYB2R151 can interact with PtrEXLB3/4, a member of the expansin gene family. It has been reported that PtEXLB3 is only highly expressed in seeds, suggesting that it might have a role in seed germination [58]. However, PdMYB2R089 was not found to interact with PtrEXLB3/4, suggesting a functional difference between PdMYB2R089 and PdMYB2R151 proteins in this respect. In fact, the subsequent results showed that PdMYB2R151, and not PdMYB2R089, could promote seed germination under both normal and drought-stress conditions.

In short, our study suggests that PdMYB2R089 and PdMYB2R151 TFs are involved in plant response to drought through multiple pathways, possibly by interacting with related proteins to regulate downstream target gene expression. However, the regulatory relationship between PdMYB2R089/151 and other proteins will need to be verified by EMSAs (electrophoresis mobility shift assays), ChIP (chromatin immunoprecipitation), and yeast two-hybrid technologies [46]. *PdMYB2R089* and *PdMYB2R151* genes were preliminarily analyzed in *A. thaliana* in terms of response to drought and are expected to be studied in poplar through gene editing technology in the future.

## 4. Materials and Methods

### 4.1. Plant Materials and Growth Conditions

In this study, the tissue-cultured plantlets of hybrid poplar NL895 (*P. deltoides* × *P. euramericana* cv. ‘Nanlin895′) were used for cloning two selected *R2R3-MYB* genes. These plantlets were grown on 1/2 Murashige & Skoog (MS) medium under a photoperiod of 16 h of light and 8 h of darkness. Wild-type (WT) *A. thaliana* Columbia ecotype (Col-0) plants were used for gene transformation and were grown at 25 °C under a 16 h light/8 h dark photoperiod.

### 4.2. Phylogenetic Analysis

The S21 and S22 subgroup R2R3-type MYB proteins of *P. trichocarpa* were downloaded from Phytozome (https://phytozome.jgi.doe.gov/, accessed on 15 April 2023), and relevant genomic data of *A. thaliana* (TAIR10) were required from the TAIR website (https://www.arabidopsis.org/index.jsp, accessed on 15 April 2023). Then, PdMYB2R089/151 proteins were used as a query in the BLASTP search of the NCBI database (https://www.ncbi.nlm.nih.gov/, accessed on 15 April 2023) to identify their most closely related proteins in monocotyledon and dicotyledon. The Interspecies phylogenetic trees for PdMYB2R089 and PdMYB2R151 were constructed using MEGA-X software (https://www.megasoftware.net/, accessed on 16 April 2023) with 1000 bootstrap replicates.

### 4.3. Structure Analysis and Protein Interaction Prediction

In the previous study, we cloned *PdMYB2R089* and *PdMYB2R151* genes from NL895 poplar and obtained their coding sequences (CDSs) by sanger sequencing [25]. ProtParam tool on the Expasy website (https://www.expasy.org/resources/protparam, accessed on 18 April 2023) was used to analyze the full-length CDSs of the target proteins, including amino acid number, amino acid types, and physicochemical properties (e.g., molecular weight, isoelectric point, instability index, aliphatic index, and grand average of hydropathicity). ProtScale (https://web.expasy.org/protscale/, accessed on 18 April 2023) was applied to analyze the hydrophilicity and hydrophobicity for the target proteins. SOPMA (https://npsa-pbil.ibcp.fr/cgi-bin/npsa_automat.pl?page=npsa_sopma.html, accessed on 18 April 2023) was utilized to predict their secondary structures. SWISSMO DEL (https://www.swissmodel.expasy.org/, accessed on 19 April 2023) was adopted to build its tertiary structure models. Meanwhile, CDD (conserved domain database, http://www.ncbi.nlm.nih.gov/Structure/cdd/wrpsb.cgi, accessed on 19 April 2023) and TMHMM-v2.0 (http://www.cbs.dtu.dk/services/TMHMM/, accessed on 19 April 2023) were employed to predict their functional domains and transmembrane helices (TMH), respectively. Finally, STRING (https://cn.string-db.org/cgi/, accessed on 25 April 2023) was used to predict protein–protein interaction for PdMYB2R089/151 proteins.

### 4.4. Subcellular Localization of the PdMYB2R089 in Poplar

The CDS of PdMYB2R089 without a stop codon was inserted into the vector p2GWF7.0 for fusion expression of the target gene and a green fluorescent protein (GFP). D53-mCherry, a nuclear localization protein with a red fluorescent label, was used as a positive control [59,60]. The subcellular localization experiment was conducted according to our previous study [27].

### 4.5. Construction of Plant Overexpression Vectors

The overexpressed vector pBI121-3HA-des was double-digested with both KpnI and SacI (New England Biolabs), and then linearized fragments were recovered. Primers for vector construction were designed by Oligo7 and listed in Appendix A. *PdMYB2R089* and *PdMYB2R151* genes were ligated to the pBI121-3HA-des vector using ClonExpress^®^ II One Step Cloning Kit (Vazyme) to construct pBI121-3HA-*PdMYB2R089*/*151* fusion vectors respectively. The reaction products were transformed into *Trelief*™ 5α Chemically Competent cells, and the monoclonal colony was selected for propagation and detected by agarose gel electrophoresis the next day. The bacterial solution with the correct stripe was sent to Tsingke (Tsingke, Beijing, China) for Sanger sequencing.

### 4.6. Genetic Transformation and Detection of Transgenic Positive Plants

Plasmid extraction of the expanded bacterial solution was performed using Anavoprep Rapid Mini Plasmid Kit (TIANGEN). 150~200 ng plasmid was transformed into 100 μL *Agrobacterium tumefaciens* strain GV3101 and then transformed into *Arabidopsis* Col-0 via agrobacterium inflorescence infection. The infected *Arabidopsis* were cultured at 25 °C under a 16 h light/8 h dark photoperiod. The first generation (T1) seeds were collected at 45 days and screened with kanamycin on 1/2 MS medium, and the screened green seedlings were transferred to soil for growth. Then, the leaves of plants were used to extract DNA as a template, and the upstream primers of the vector (Appendix A) and downstream primers of the target gene were used for PCR validation of the positive plant seedlings. The same method was used to continue screening the T2 generation transgenic seeds until we obtained genetically stable transgenic lines.

### 4.7. Drought Stress Management and Seed Germination Phenotype Analysis

Seed germination and seed root growth of transgenic *Arabidopsis* under drought stress were mainly performed on 1/2 MS agar plates supplemented with different concentrations of PEG6000 (0%, 5%, and 10%). The disinfected wild-type and *PdMYB2R089*/*151*-OE seeds of *Arabidopsis* were seeded on 1/2 MS plates with different drought-treatment concentrations. WT (36 grains) and transgenic seeds (36 grains) were seeded on each plate with two replicates under each treatment. To break the seed dormancy, seeds were incubated at 4 °C for 2 days and then placed under normal growth conditions to record seed germination.

### 4.8. Plant Fresh Weight and Phenotype Analysis

WT and transgenic seeds were seeded on 1/2 MS medium without PEG for about 10 days, then WT (6 lines) and transgenic *Arabidopsis* (6 lines) transferred to 1/2 MS plates with different PEG concentrations for 6 days, respectively. The root growth state was recorded every 2 days, and the total FW and the roots-FW were weighed on the 6th day, with 6 lines for each treatment and three replicates for each plant.

### 4.9. Stomatal Variation and Life Cycle of Arabidopsis

Three leaves of WT and transgenic *Arabidopsis* potted seedlings were randomly selected under normal water conditions, and their stomata sizes were observed by microscope with 3 replicates in each field. The potted seedlings were treated with 7 days of drought and 3 days of rehydration. The changes in stomata were observed, and the pore size was statistically analyzed. In addition, this study also investigated whether these three genes affect the flowering and fruiting cycles of plants. WT and transgenic *Arabidopsis* with consistent growth were transplanted into soil and cultured in the same environment to observe flowering and fruiting.

### 4.10. Determination of Physiological Indicators Related to Drought

Coomassie brilliant blue G-250 was used to detect the contents of soluble protein [61]. We weighed 0.5 g of fresh plant leaf samples, added 6 mL phosphoric acid buffer (0.1 mol/L, pH = 7.0) for grinding, centrifuged this, and took the supernatant to obtain the MDA extract liquid. A total of 1 mL MDA was used to configure the reaction solution, and supernatant absorbances were detected at 532 and 600 nm to estimate the MDA content. POD enzyme activity was determined by the guaiacol method, and absorbance was measured by spectrophotometer at 470 nm. The absorbance was recorded once per minute, and the enzyme activity was expressed by the change in absorbance per minute [62].

## 5. Conclusions

In this study, we analyzed the gene structure, interspecific evolutionary relationship, and drought responsive function of *PdMYB2R089* and *PdMYB2R151*. The bioinformatic analysis results showed that the two PdMYB2R proteins had high thermal stability and typical MYB conserved domains, which implied similar functions. However, the protein–protein interaction prediction suggested that the function of the two PdMYB2R proteins might have minor differences in some respects. The phylogenetic analysis showed that *PdMYB2R089*/*151* was most closely related to the homologous genes in four *Populus* species, followed by the *Salix* species. The *PdMYB2R089*-OE transgenic plants had a high growth performance in terms of their above-ground parts, roots, and fresh weight under drought stress. *PdMYB2R089* and *PdMYB2R151* could induce stomatal closure rapidly in a dewatering environment, which can enhance drought tolerance by reducing water loss in plants. *PdMYB2R089* and *PdMYB2R151* could reduce the accumulation of drought-induced MDA by enhancing POD enzyme activity. Moreover, *PdMYB2R151* could also promote seed germination under drought stress, but *PdMYB2R089* could not. On the other hand, *PdMYB2R089*/*151* could significantly accelerate flowering and fruiting and shorten the growth cycle in transgenic *Arabidopsis* plants. In conclusion, the results indicate that *PdMYB2R089* and *PdMYB2R151* could positively regulate the process of plant responses to drought stress. However, the regulatory roles of poplar *R2R3-MYB* genes in drought regulation networks are rarely documented. The regulatory relationship between *PdMYB2R089*/*151* and other TFs or downstream drought-related genes might be further verified by ChIP and Y1H/Y2H assays.

## Figures and Tables

**Figure 1 ijms-24-13466-f001:**
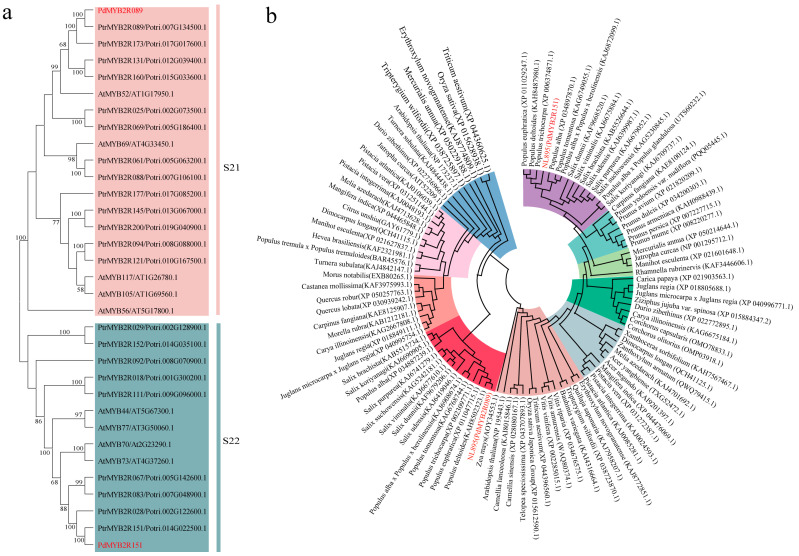
Evolution tree of *PdMYB2R089* and *PdMYB2R151* genes. (**a**) shows the phylogenetic trees of the gene members in the S21 and S22 subgroups of poplar and *A. thaliana*. (**b**) shows the evolutionary tree of the *PdMYB2R089*/*151* genes and their homologous genes in other species. The two target genes are highlighted in red.

**Figure 2 ijms-24-13466-f002:**
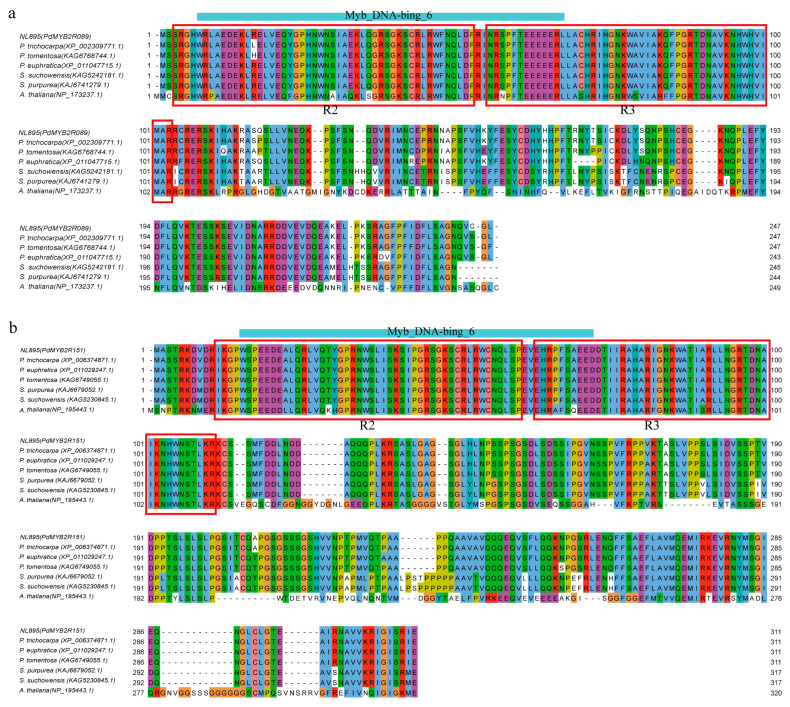
Conserved functional domains in PdMYB2R089/151 proteins. Subfigures (**a**,**b**): the sequence alignment of NL895 PdMYB2R089/151 and the homologous proteins of *Populus*, *Salix*, and *A. thaliana*. The blue bars represent the DNA binding domains.

**Figure 3 ijms-24-13466-f003:**
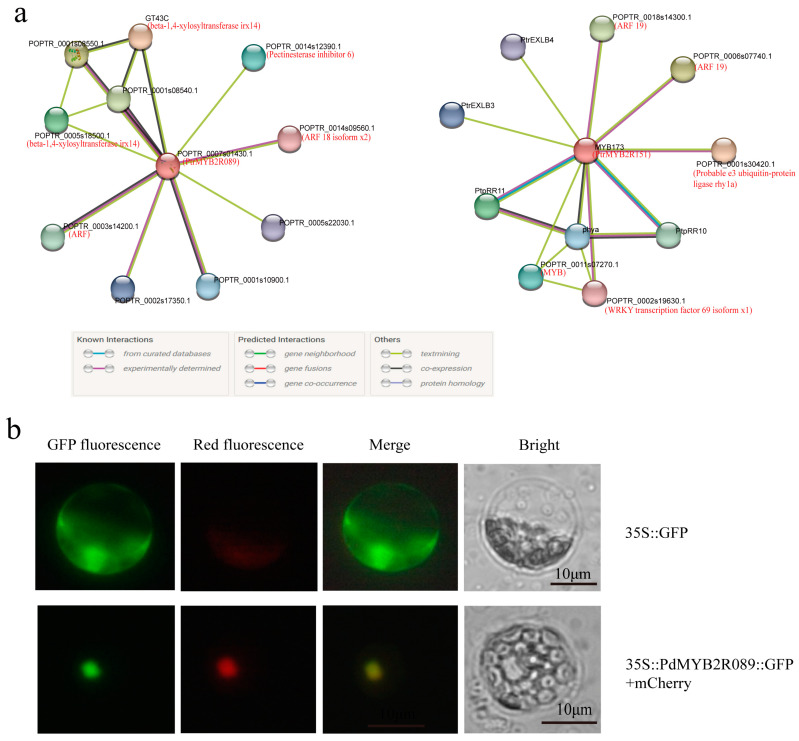
Protein–Protein interaction and subcellular localization. (**a**): prediction of protein–protein interaction. The network nodes represent the proteins: the splice isoforms or post-translational modifications are collapsed, i.e., each node represents all the proteins produced by a single, protein-coding gene locus. Information about the interacting proteins is shown in Appendix A. (**b**): subcellular localization of PdMYB2R089. mCherry-D53: a red and nuclear-localized marker. Scale bar: 10 μm.

**Figure 4 ijms-24-13466-f004:**
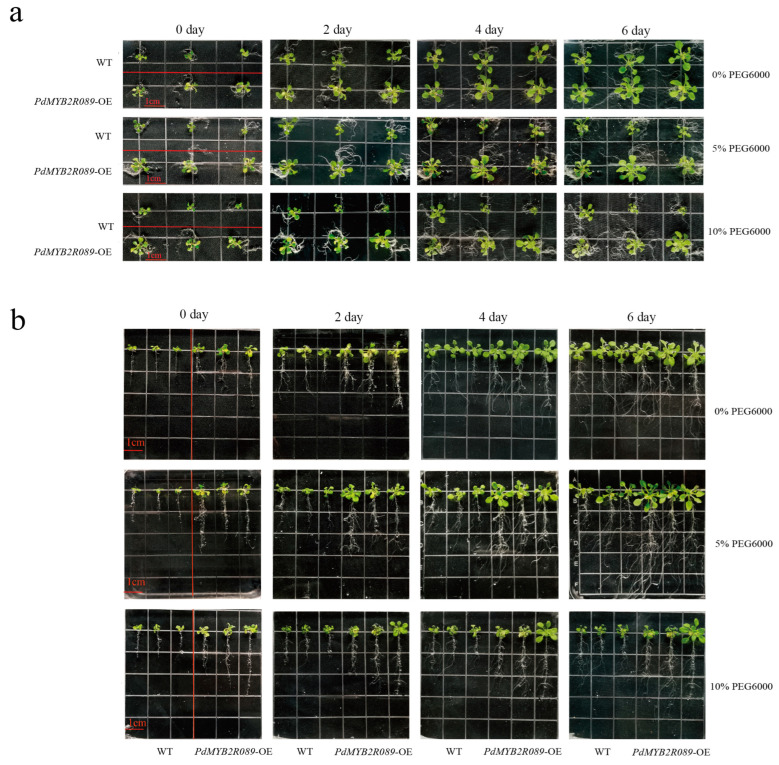
Growth phenotypes of *PdMYB2R089*-OE transgenic plants in the above-ground and root systems. (**a**) shows the above-ground growth phenotypes of WT (three lines) and *PdMYB2R089*-OE transgenic plants (three lines) under different drought stress. (**b**) shows the root growth phenotypes of WT (three lines) and *PdMYB2R089*-OE (three lines) under different drought stress. The seedlings grown on the normal medium for 10 days were transferred to a medium containing different concentrations of PEG6000 for drought treatment. The size of each small square is about 1.3 cm × 1.3 cm. The red line indicates the separation between WT and *PdMYB2R089*-OE *Arabidopsis*.

**Figure 5 ijms-24-13466-f005:**
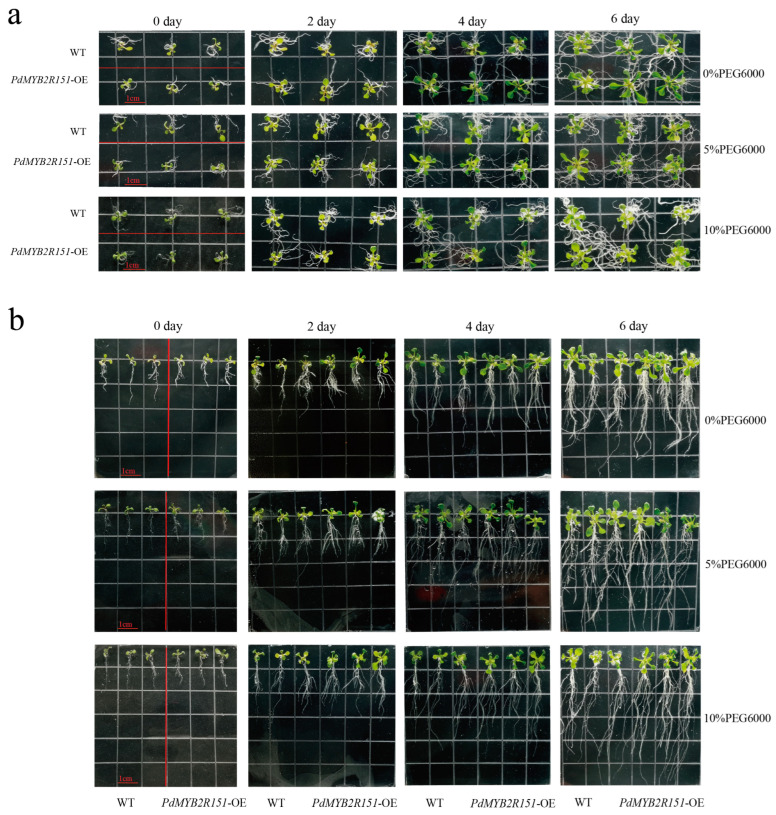
Growth phenotypes of WT and *PdMYB2R151*-OE plants in the above-ground and root systems. (**a**) shows above-ground growth phenotypes of WT (3 lines) and *PdMYB2R151*-OE transgenic plants (3 lines) under different drought stress. (**b**) shows root growth phenotypes of WT (3 lines) and *PdMYB2R151*-OE (3 lines) under different drought stress. The size of each small square is about 1.3 cm × 1.3 cm. The red line indicates the separation between WT and *PdMYB2R089*-OE *Arabidopsis*.

**Figure 6 ijms-24-13466-f006:**
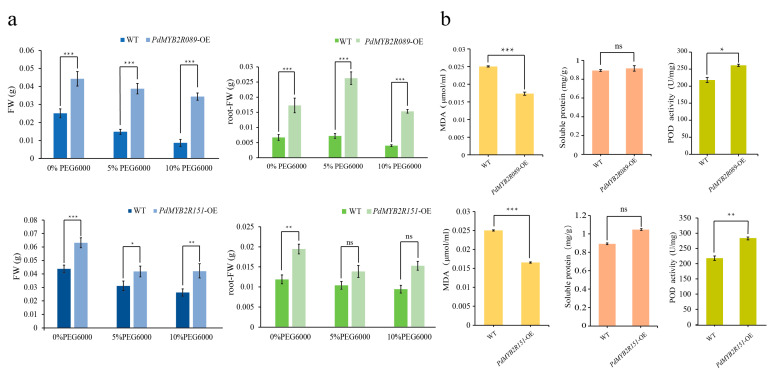
Statistics of various indicators of WT and *PdMYB2R089/151*-OE transgenic plants under drought stress. (**a**) shows the total FW and root-FW statistics of *Arabidopsis* under different drought concentrations. There were six seedlings for each treatment. (**b**) shows the statistics of MDA content, soluble protein, and POD enzyme activity in leaves after 7 days of water shortage. There were three repetitions of each treatment. * indicates *p* < 0.05; ** indicates *p* < 0.01; *** indicates *p* < 0.001; ns indicates not significant.

**Figure 7 ijms-24-13466-f007:**
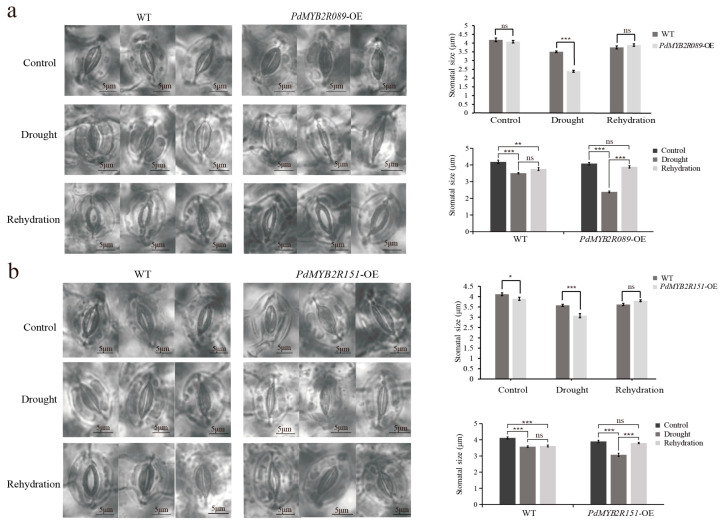
Statistics of stomatal pore size of WT and transgenic *Arabidopsis* after drought and rehydration treatment. (**a**) Changes in stomatal pore size of WT and *PdMYB2R089*-OE *Arabidopsis* after control, drought and rehydration. (**b**) Changes in stomatal pore size of WT and *PdMYB2R151*-OE *Arabidopsis* after control, drought and rehydration. There are three repeated views of each treatment in the picture; scale bar: 5 μm. Statistical graph: * indicates *p* < 0.05; ** indicates *p* < 0.01; *** indicates *p* < 0.001; ns indicates not significant.

**Figure 8 ijms-24-13466-f008:**
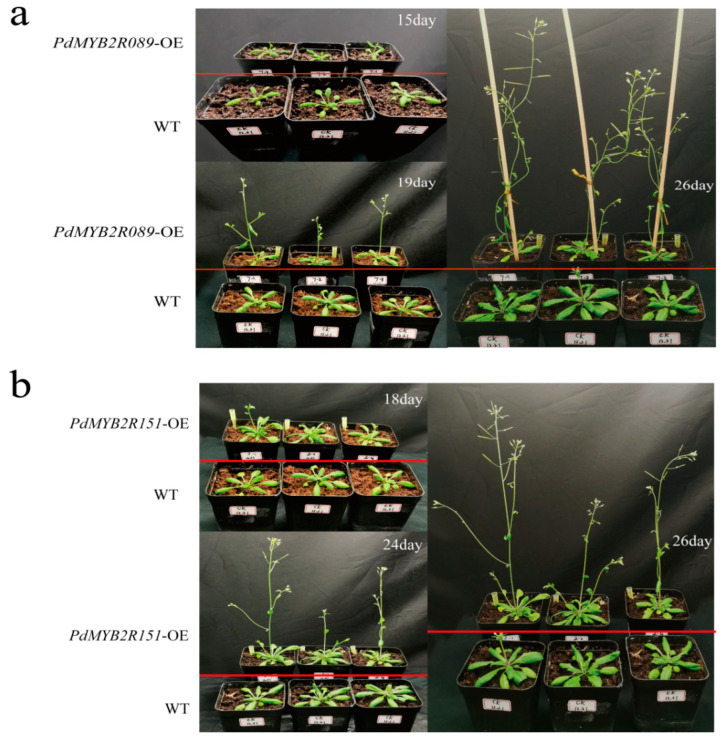
Flowering and fruiting time of WT and transgenic plants. (**a**) Flowering and fruiting time of WT and *PdMYB2R089*-OE *Arabidopsis*. (**b**) Flowering and fruiting time of WT and *PdMYB2R089*-OE *Arabidopsis*.

## Data Availability

All datasets presented in this study are included in the article’s Appendix A.

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
