# Peer review of "The Over-Expression of Two *R2R3-MYB* Genes, *PdMYB2R089* and *PdMYB2R151*, Increases the Drought-Resistant Capacity of Transgenic *Arabidopsis"

_ijms, 2023, doi:10.3390/ijms241713466_

Round 1
Reviewer 1 Report
Zhang et al “Over-expression of two R2R3-MYB genes, PdMYB2R089 and PdMYB2R151……..” investigated the roles of those Populus genes in drought by overexpressing in Arabidopsis.
Comments.
Authors need to explain well what is Pd, Pto, Ptr terminology in introduction. How many total species of Populus are known?
Give some background of other subgroups. Author abruptly started with S21, S22
Figure1a: authors used Pd, Ptr, why not Pto as well
Figure 2b,c. what is difference between 2B and 2C. Need more explanation in legend as well. Why PdMYB2R089 and PdMYB2R151 are not used in same dendrogram.
Figure 2b and 2c. both Populus gene cluster with Salix. What is Salix and relationship with Populus.
Line 95: I never heard inflorescence impregnation. I guess they mean infiltration.
Line 149: where is MYB DNA binding domain in figure 2
Line 151: need more explanation what is two tandem incomplete replicates (R2 and R3)
Effect of both gene in transgenic Arabidopsis are similar. To avoid redundancy in results, authors can combine 2.4 and 2.5 into one section and move one set of figures in supplementary section. This version looks dissertation format.
Need explanation of statistical analysis (both in methods and figure legends) and how many plants per treatment were used.
Authors need minor improvement in writing
Reviewer 2 Report
The manuscript by Zhang et al. presents an analysis of the role of poplar R2R3-MYB S21 and S22 subgroup genes in the drought response. In their previous research (ref. 24), the authors analyzed the expression of 10 poplar R2R3-MYB genes, including PdMYB2R089 and PdMYB2R151, the homologues of the Arabidopsis AtMYB52 and AtMYB73 genes, presenting different R2R3-MYB subgroups. In the current manuscript, the authors 1) used bioinformatics tools to analyze phylogeny, secondary and tertiary structure, predict the physicochemical properties and functional domains of PdMYB2R089 and PdMYB2R151 proteins, as well as their interactions with the other proteins; 2) obtained the transgenic Arabidopsis lines, overexpressing PdMYB2R089 and PdMYB2R151, and characterized various drought-adaptive features of these transgenic plants.
Comments:
1) Line 18: “Their regulatory mechanisms and functions about drought stress were studied through the Arabidopsis floral dip transformation.” I do not think that this method allows to study gene regulation. Regulatory mechanisms can only be assumed based on the morphological/biochemical properties of transgenic plants.
2) Line 97: “The research regarding the regulatory mechanisms of PdMYB2R089 and PdMYB2R151 genes under drought stress would be significant for the functional study of the poplar R2R3-MYB S21 and S22 subgroup genes.” The same issue; current study does not provide information on the regulatory mechanisms. Did the authors mean further perspectives?
3) Line 194: “Under normal conditions, the above-ground growth of PdMYB2R089-OE Arabidopsis was similar to the wild type (WT).” I cannot agree with this conclusion, as in the panels corresponding to “Day 0” of Figure 4A, the above-ground parts of plants are larger in PdMYB2R089-OE Arabidopsis.
4) Lines 238 and 323: In Sections 2.4 and 2.5 the authors end the description of transgenic Arabidopsis with the special subsections, summarizing the conclusions related to each section. In my opinion, these sub-conclusions should be transferred from “Results” to the appropriate sections of the manuscript (“Discussion” and “Conclusion”).
5) Figures and 5 and 7: Please provide the description of the grayscale images at the right-hand side of panel C. These stomatal images should include some labels or arrowheads indicating features that the reader should pay attention to. Why does each panel include three images, are these three replicates? The scale bars are also missing.
6) Line 280: In a case of PdMYB2R151-OE, the above-ground growth at point 0 was similar to that in WT (unlike PdMYB2R089-OE).
7) Line 286: “… transgenic plants had more lateral roots and longer root length than WT.” I would say, slightly longer roots than WT (as compared with PdMYB2R089-OE).
8) Lines 214 and 296: In the sections which present an analysis of stomatal movement, the authors compare stomatal size after drought treatment and dehydration with the control (separately for WT and transgenic plants). For example: “The stomatal conductance of transgenic Arabidopsis was restored after rehydration treatment, while that of WT plants was not fully restored to that before drought-stress treatment”.
In Figures 5 and 7 (C) the stomatal size is statistically compared between WT and OE plants. I think, the figures should provide information on statistically significant differences between the “control” “drought” and “rehydration” conditions to provide a statistical basis for the conclusions in this section.
9) Line 304: “Overexpressed PdMYB2R151 reduced drought damage to plants by antioxidant stress pathway”. In my opinion, this is an over-inference that could be suggested by the results, but a detailed molecular analysis is not presented in the manuscript.
10) Line 332: “In addition, we should note that the growth regulatory phenotype of the two genes still appeared minor difference.” The meaning of this sentence is unclear.
11) Line 362: “we found a close relationship between poplar…” I would say: “Our results suggest a close relationship…”
12) Line 407: “Our result showed that PdMYB2R151 would promote root growth and lateral root development in an arid environment so that the transgenic plants could absorb more water from the medium to maintain normal growth and development of the plants.” This conclusion is not supported by the results, because the images do not show significant lateral root growth of the PdMYB2R151-OE plants, and the difference between the root FW between PdMYB2R151-OE and WT plants is not statistically significant under drought stress (Figure 7b).
13) Line 475: Please provide more information about the nuclear localization of D53, I did not find it in ref 50.
Minor corrections:
1) Line 127: I suggest not to use the shortened name in the title of the section. Should be: “PdMYB2R089 and PdMYB2R151 proteins”
2) Line 271: “Members of the R2R3-MYB gene…” Gene family?
3) Figure 7 (g): Please remove the label “105%” from the vertical axis.
4) Line 349: “Protein-protein interaction prediction showed that PdMYB2R089/151 might interact with PRR protein influencing plant flowering.” Please provide the reference here.
5) Line 380: “These results suggest that PdMYB2R089/151, as the gene member of S21/S22, may interact…” These are two different genes from different subgroups.
6) Line 403: “receptor coding gene PYL8”. Are these interactions related to receptor, or to the gene?
Round 2
Reviewer 1 Report
Authors addressed my all concerns and comments and improved manuscript accordingly.
Thanks
Author Response
Thank you very much for your inspiring comments on our manuscript.
Reviewer 2 Report
During the revision, the authors carefully addressed all my comments. I have only one question left.
Line 288 (Figure 7, legend): “Different letters (a, b) in the statistical graph indicate significant differences (P < 0.05), while the same letters indicate no significant differences (P > 0.05).” How can we judge the statistical significance when the same letters stand for both significant and non-significant comparisons?
Author Response
Dear reviewers,
Thank you for your comments. To better represent the significance between "Control", "Drought", and "Rehydration", we have modified the image to show the significance with asterisks and lines (Figure 7).